# Prediction of Satellite Shadowing in Smart Cities with Application to IoT

**DOI:** 10.3390/s20020475

**Published:** 2020-01-14

**Authors:** Susana Hornillo-Mellado, Rubén Martín-Clemente, Vicente Baena-Lecuyer

**Affiliations:** 1Department of Signal Theory and Communications, University of Sevilla, 41092 Sevilla, Spain; 2Department of Electronic Engineering, University of Sevilla, 41092 Sevilla, Spain; v_baena@us.es

**Keywords:** IoT, satellite communications, reliable broadcast, smart cities, satcom on-the-move

## Abstract

The combination of satellite direct reception and terrestrial 5G infrastructure is essential to guarantee coverage in satellite based-Internet of Things, mainly in smart cities where buildings can cause high power losses. In this paper, we propose an accurate and fast graphical method for predicting the satellite coverage in urban areas and *SatCom on-the-move* scenarios. The aim is to provide information that could be useful in the IoT network planning process, e.g., in the decision of how many terrestrial repeaters are really needed and where they should be placed. Experiments show that the shadowed areas predicted by the method correspond almost perfectly with experimental data measured from an Eutelsat satellite in the urban area of Barcelona.

## 1. Introduction

5G networks are expected to provide connectivity to everyone, everywhere, at anytime with increased capacity and higher user data rates than today’s capabilities [1]. This will serve especially to establish the Internet of Things (IoT), which is aimed at collecting and sending data with different purposes and over different usage scenarios, such as enhanced Mobile Broadband (eMBB), Ultra-Reliable and Low Latency Communications (URLLC), and massive Machine Type Communications (mMTC) [2,3,4].

As is well-known, terrestrial-only networks cannot ensure a proper access to Internet and other communication services in remote areas or where the infrastructure is damaged due to natural extreme events, just to cite a few examples. Smart cities, in addition, will bring huge demands for broadcasting infrastructures with broadband connectivity and reliable emergency communication networks. All these services will benefit from the combination of a variety of communication technologies, including non-terrestrial platforms. In consequence, a great effort is being made by researchers and the 5G industry to develop new standards of interworking among different access technologies, specially focusing on the integration of different terrestrial and satellite networks, aiming at exploiting the possible synergies between them [5]. As a further benefit, it is relevant to mention that, in addition to providing data links in those areas where terrestrial infrastructures are not available, satellites are also recognized as the more efficient option for certain machine type communications, information dissemination, broadcast, as well as for some delay tolerant services [6,7]. Moreover, the satellite bandwidth cost has fallen dramatically since the appearance of the first High-Throughput Satellites (HTS). It can therefore be said, in conclusion, that satellites are a reliable and cost-effective complement to terrestrial infrastructure for delivering broadband communications [8].

According to the European Commission Horizon 2020 5G Public Private Partnership Phase 2 project “SaT5G” (Satellite and Terrestrial Network for 5G) [9], the analysis of the on-going 5G specification shows two main options for the satellite radio access network (RAN) in the future 5G architecture (see Figure 1):Direct access: satellite-capable UE (*User Equipment*) has a direct access to the 5G network through a satellite link.Indirect access: UE accesses to RAN that is connected to the 5G core through a satellite link.

The indirect access is particularly relevant for Internet of Things (IoT) applications. This is easy to understand if we think that most IoT end-devices lack the ability to communicate directly with a satellite, as it would dramatically increase their cost and power consumption. Just to cite one illustrative example, an eye-catching satellite backhaul-based IoT network is discussed in [5,10]. This contribution develops the idea of using low altitude drones and unmanned aerial vehicles (UAVs) as IoT gateways that collect the data provided by all the short-range IoT devices spread in the area. Satellites are then used to connect the UAVs to the main Internet backbone, which can be seen as a paradigmatic “comms on the move” application [11]. Note the pilot experiment carried out in the Rwandan capital of Kigali, where the Inmarsat consortium is deploying a network of LoRa-based IoT devices (such as sensors in water resources) [12], using exclusively satellite communication as the backhaul. In addition to illustrating the concept, the project will serve as a blueprint for the quick development of smart cities in areas where the terrestrial 5G infrastructures are not mature enough or need to be complemented (e.g., in temporary deployments for massive public events).

However, before we can ensure that IoT systems perform to their full potential over satellite networks, some serious challenges have to be addressed. On the one hand, IoT protocols (Zigbee, LoRa, NB-IoT, and Sigfox *…*) [4] are not well-suited to the considerable delays in the satellite-earth radiolinks, and consequently, new approaches or amendments of the existing ones are required, whatever the mode of access (see e.g., [13,14,15]). On the other hand, most of the typical deployment scenarios for eMBB, mMTC, and URLLC analyzed in the 3rd Generation Partnership Project (3GPP) technical report [7] are intended to be implemented in smart cities, where the goal is to maintain continuous and ubiquitous coverage with high traffic loads. In this context, satellites could help to alleviate the traffic congestion of terrestrial 5G infrastructure during peak hours by broadcasting large amounts of delay non-sensitive data, so this is a market in which satellite providers are increasingly interested. However, in urban environments the satellite signal is often blocked by the high density of buildings, resulting in a considerable attenuation of the received signal. The situation becomes even more critical in the case of mobile terminals, which are encountered, e.g., on SatCom On-The-Move applications [11,16,17], for the environment changes as the terminal moves [18]. In this context, the new generation of autonomous vehicles raises new challenges, especially with regard to the safety of their use [19]. A stand-alone satellite backhauling can be envisaged for complementing existent terrestrial connectivity in order to ensure the reliability of communications [1].

To prevent link interruptions and keep the services within acceptable limits of quality, deep interleaving mechanisms (time diversity) and several constellation arrangements, including GEO (geostationary) or non-GEO satellites (satellite diversity), can be used. Space/polarization diversity based on Multiple Input Multiple Output (MIMO) techniques can also improve the QoS (Quality of Service) [20]. Finally, in the worst cases, in urban areas where the channel is hardest, we will have no choice but to use broadcast systems based on hybrid terrestrial satellite networks [21]. A seamless hand-off mechanism will then be required.

To the best of our knowledge, many of the previous issues are overlooked in the current research efforts to developing satellite-based IoT networks, especially with regard to GEO satellites. Frequently, only the challenges caused by the high communication latency between the ground terminals and the satellite, and the collisions in the data transmission, are addressed in the literature. By contrast, it is usually assumed by default that the communication with the satellite is reliable and works without interruptions in the data flow, which is questionable and ultimately limits the validity of the results except in rural areas. The choice of non-GEO orbits (or GEO/non-GEO combinations) with sufficiently large constellations allows solving, to a great extent, the problem of accessibility in urban areas (take as an example the recent Japanese QZSS (Quasi-Zenith Satellite System), which combines Quasi-Zenith orbits (QZO) with GEO improving significantly the GPS accuracy [22]). However, GEO satellites remain essential for broadcasting applications and managing high volumes of data over large areas. Moreover, GEO satellites can operate with a variety of ground equipment ranging from very large fixed gateway stations down to mobile terminals. These features are even more relevant in emergency situations where the reliability of communications becomes critical. In fact, GEO satellites are a key piece in the International Charter “Space and Major Disasters” [23].

In this regard, to carry out the planning of the network, we first require reliable propagation models that are simultaneously easy to implement and computationally efficient, so that they may be even implemented in some terminals and the IoT gateways. In this paper, we address this issue and propose simple geometry-based physical model for estimating satellite coverage areas in urban scenarios. In particular, we use detailed cartographic information to predict the “shadow zones”, that is, those areas where the satellite Line-Of-Sight from is blocked by buildings. The results are validated with real data measured in the urban area of Barcelona from a geostationary satellite working in the S-band.

The rest of the article is structured as follows. Section 2 reviews the basics of satellite-based IoT networks and radio propagation models. Section 3 presents the basis of the proposed approach for determining the shadowed areas. Experiments validating the method on real data are reported in Section 4. Some discussion on the implications for the design of IoT communication protocols is given in Section 5. The concluding remarks of Section 6 bring the paper to an end.

## 2. Background

In this Section, after a brief overview of the available technologies, we present and discuss the different techniques for predicting the coverage of the satellite radiolink in urban areas.

### 2.1. Satellites for IoT

In the context of 5G technology, the integration of terrestrial and satellite systems can be considered through Geostationary Earth Orbit (GEO), Medium Earth Orbit (MEO), or Low Earth Orbit (LEO). The advantages and disadvantages of each one of them for IoT communications have been thoroughly debated in the literature (see, e.g., [15,17]).

GEO satellites provide high-bandwidth and high-reliability. Furthermore, a single GEO satellite can broadcast communications over wide areas, including remote rural zones where terrestrial infrastructure is unavailable. However, they are not suitable for applications that require a low latency, as the altitude of GEO satellites above sea level is about 36,000 km, and therefore the propagation delay from the satellite to the earth is budgeted at approximately 125 ms.LEO and MEO satellites can deliver delay-sensitive services due to their lower orbit altitude, and the signal losses in the radio link power are also smaller for the same reason. Unlike GEO satellites, which remain static to the ground stations, LEO and MEO satellites move at a higher speed, completing their orbits in a short time (about 100 min for LEO satellites). Therefore, they can avoid obstacles near the terminal that might otherwise hinder the communication. The downside is that a constellation of many satellites is required to ensure the global coverage of the Earth surface, increasing the complexity of the system. In addition, handover mechanisms are required as the satellite that disappears over the horizon must be seamlessly replaced with other to maintain the communication.

The combination of LEO/MEO constellations and multi-layer missions including LEO/MEO/GEO satellites and HPAS (High Altitude Platform Systems) are also considered in the next generation segment for integration within 5G [9].

To cite only a few commercial options, Eutelsat (European Organisation of Telecommunications by Satellite S.A.) offers GEO-based IoT services with transfer speeds up to 1 Mbps in the satellite-to-ground communication and up to 128 kbps in the opposite direction. Inmarsat (International Maritime Satellite (consortium)) has also announced its intention to provide satellite connectivity to LoRa Wide Area Networks (LoRaWANs) (see, e.g., [13] for a discussion about how to enable LoRaWAN services in GEO-based satellites). Both of them provide M2M services for applications with stringent timing requirements, normally associated to the massive synchronization of sensors and other devices. On the other hand, Iridium, which is a LEO satellite system, has recently started providing data communication links for IoT devices. ORBCOMM and Globalstar also offer LEO satellites to provide M2M communications or remote data collection. Other providers, such as OneWeb or SpaceX, will begin offering similar services in the near future. Finally, the next generation MEO constellation O3b POWER rapidly expandable and highly scalable is expected to begin its launch in 2021 and will be able to offer several terabits of performance worldwide [24].

### 2.2. Satellite Coverage Estimation

The wireless link between the satellite and the receiver is considerably fragile due to the small transmitted powers. The problem is specially acute when the receiver is moving, as, in this case, multipath interference may be significant, or the signal may be blocked by buildings or trees, depending on the receiver’s location [21]. It is, therefore, unlikely that there will be a line-of-sight connection all the time.

It is generally a demanding task to predict the coverage of a satellite radio link in urban areas, and a number of approaches have been developed to evaluate the shadowing effects due to buildings on land mobile satellite (LMS) channel [25,26,27,28,29,30,31]. In general, we can classify the different techniques into four categories:1.Deterministic ray-tracing coverage approaches.2.Techniques based on Masking Functions.3.Statistical approaches, which describe the channel in terms of statistical distributions.4.Empirical models, which fit mathematical expressions to measured attenuation data.

First, satellite coverage estimation can be carried out using traditional ray-tracing techniques, where each ray is associated with an electromagnetic wave and the total field at a point is obtained by summing the fields of all rays through the point. High accuracy can be obtained by considering not only the incident and reflected plane waves as well as the effect of the rays diffracted at the boundary surfaces [25,32]. The influence of buildings, trees, and other obstacles to the received signal can be easily incorporated to the model in this way. Unfortunately, the computational burden of ray-tracing techniques make them practically unfeasible for the prediction of the satellite coverage areas in real urban scenarios.

Alternatively, in [27,28,33], an approximate methodology based on street-masking functions (MKFs) for urban areas is presented. The MKFs indicate the azimuth and elevation angles for which a satellite is visible from the user terminal’s location. Functions of this type have often been obtained by means of photogrametric studies and ray-tracing. Alternatively, realistic environments can be also modeled by creating virtual scenarios using observed distributions of building heights, widths, spacings, etc. [21]. A simplified version of the MKF concept, which yields an approximate evaluation of the availability of the satellite signal, can be derived when the contribution of the reflected and diffracted components is disregarded. In this case, MKFs can be calculated according to basic geometrical considerations for a limited number of standard scenarios, where only a minimal knowledge of the urban structure (generic street widths and the average building height) is required. For example, in simple situations (e.g., street canyons, street crossings, T-junctions, or single walls) the MKFs are defined by generic relationships such as the following one [33]:(1)$$\theta =\arctan\left(h/\sqrt{{\left(\frac{w}{2}\right)}^{2}\left(\frac{1}{\tan^{2}\varphi }+1\right)}\right)$$
where θ and φ are the elevation and azimuth angles relative to the position of the satellite, *h* is the average building height, *w* is average street width in the urban area, and the mobile terminal is supposed to be located in the middle of the street. Depending on the particular configuration, reflected versions of this function are also used to obtain the MKF. For example, some relevant MKFs are illustrated in Figures 10 and 11, corresponding to generic street configurations found in the city of Barcelona. Shaded areas represent the “forbidden” zones, that is, the elevation and azimuth angles for which the link can not be completed.

Statistical techniques, as a third approach, characterize the fading of the received signal, i.e., the changes in the received signal amplitude, by probability distributions. For mathematical convenience, narrowband fading is characterized by two main effects [18,34]: low/very slow fading, due to large-scale features of the environment, which may induce shadowing (slow fading) or blockage (very slow fading) of the direct signal, and fast fading, which is superimposed to slow (and very slow) fading and is due to multipath effects occurring in objects near the receiver. For example, in Figure 2, a fragment of the signal level received during our experiments by a mobile terminal from a geostationary satellite (S-band) is shown, where the fast/slow and very slow variations are represented. A popular statistical characterization of these phenomena, proposed by Loo [35], considers that the phasor seen by the receiver is of the form
$$R\;\exp\left(j\theta \right)=\sigma \;\exp\left(j\phi_{1}\right)+Y\;\exp\left(j\phi_{2}\right)$$
where $\phi_{1}$ and $\phi_{2}$ are uniformly distributed between 0 and 2π, σ is log-normally distributed and *Y* has a Rayleigh distribution. In this model, σ represents the dominant component, mainly affected by shadowing, and $Y\;\exp(j\phi_{2})$ is a complex quantity that models the multipath random contribution. The distribution of *R*, referred to as the Loo distribution [35], is given by
$$f_{R}\left(r\right)\approx \left\{\begin{array}{cc} \frac{1}{r\sqrt{2\pi d_{0}}}\exp\left[-{\left(\ln r-\mu \right)}^{2}/2d_{0}\right] & \mathrm{for}>>\sqrt{b_{0}}\\ \frac{r}{b_{0}}\exp\left(-r^{2}/2b_{0}\right) & \mathrm{for}<<\sqrt{b_{0}} \end{array}\right.,$$
where $b_{0}$ represents the average scattered power due to multipath, and μ and $d_{0}$ are the mean and variance, respectively, of the lognormal variable. On this regard, the local mean around which the total fading oscillates in the area of interest is calculated by Friis formula or empirical models based on field measurements. Loo is useful for describing a variety of situations, ranging from clear LOS scenarios to moderate direct signal blockage in trees. Actually, this model is a variant of classical Rician fading, where the lognormal shadowing affects the direct line-of-sight (LOS) component only, whereas a constant average power is assumed for the scattering contribution. It was primarily designed for rural environments, where the line-of-sight signal component is available at the receiver most of the time. For the NLOS situation, other approaches, e.g., see [36], propose a Rayleigh distribution where the mean received power is log-normally distributed. As another popular alternative, and on the basis of different considerations, the received signal strength at a point has also been described by Suzuki [37] as
R=σY
where σ represents the slow fading and is a lognormal random variable, and *Y* denotes a Rayleigh variable independent of σ. The distribution of *R* is called the Suzuki distribution, and is widely accepted for NLOS urban channels (see [38] for an interesting discussion on the physical interpretation of this model). Complementarily, when the terrestrial device has an unobstructed line of sight with the satellite, *Y* is better described with a Rice distribution than with a Rayleigh distribution, which has been studied in [39,40]. Specifically, the Corazza distribution [39] is valid for every kind of environment (urban, suburban, or rural) giving the proper values to the parameters.

In any case, as shown in Figure 2, the field strength varies with position in the same way as reflections, diffractions, and scattering in buildings. As a result, the parameters (shape and location) of the previous statistical distributions can only be maintained constant over short distances (a few dozens of wavelengths). The nonstationarity of the signal statistics is therefore more appropriately characterized by *multi-state models*. One of the most accepted strategies is to consider that in built-up areas the mobile terminal can only be in two possible states due to building blockage: GOOD and BAD, corresponding to LOS and NLOS [29,33,34,36,41]. Actually, the number of states considered for the Markov chain can vary, according to the number of possible shadowing states of the signal (e.g., “line-of-sight conditions”, “moderate shadowing conditions”, and “deep shadowing conditions” [42]), but it is the two-state Markov chain the one recommended by the ITU-R [33]. The global fade statistical distribution can therefore be written as
$$f_{R}\left(r\right)=p\left(B\right)\;f_{G}\left(r\right)+\left(1-p\left(B\right)\right)\;f_{B}\left(r\right)$$
where $f_{G}\left(r\right)$ is the fade probability density function (pdf) in GOOD areas, $f_{B}\left(r\right)$ is the fade pdf in BAD situations, and p(B) is the percentage of GOOD areas, often estimated by inspection. The distributions associated to each state (e.g., Rice for GOOD, Suzuki for BAD, Loo, etc.) are characterized from extensive measurement campaigns or, even, by ray-tracing considerations in simplified environments such as street canyons and crossings.

The transition between states is usually governed by a Markov model and fade probabilities are calculated for average cities urban areas are modeled by a number of standard configurations, where buildings are considered as blocks of a specific height or obtained from a random generator database with parameters that have been measured in real environments. For example, the approach proposed in [29] assumes simple urban area configurations such as street canyons and crossings and performs elementary ON/OFF (LOS/NLOS) ray-tracing to define the two possible states of a Markov chain. The influence of both elevation and street orientation angles are also taken into account. In [18], the terminal’s driving direction is considered using an image-based state estimation method. Hemispheric images of the environment were obtained from a fisheye camera pointing towards the sky. Each image is then characterized by sky and buildings regions that are identified to LOS/NLOS situations.

Finally, empirical models are derived by fitting curves to measured data, and give rough approximations of the path loss while requiring a low computational cost [43]. They are, however, of limited accuracy when the actual data is far from the data from which the model was developed.

## 3. Geometrical LOS-NLOS Approach

There is a clear trade-off between accuracy and complexity in the methods reviewed in the previous Section. For example, statistical methods cannot provide an exact expression for the electromagnetic field at a point. Instead, they give the probability of exceeding or not a certain level of signal. Exact ray-tracing methods, on the other hand, are computationally infeasible for analyzing the performance of satellite-based IoT networks and protocols. Searching for a middle ground, a popular approach in SatCom On-the-Move applications consists of calculating the satellite coverage from hemispheric images of the city obtained by using fisheye cameras, which is also computationally demanding [16,18]. Alternatively, aiming at suppressing the need for costly fisheye images, we propose to combine a geometry-based physical model with detailed cartographic information to obtain a more accurate estimation of the shadowed areas (NLOS) where the signal from the satellite is blocked by buildings. The results are validated with real data measured in the urban area of Barcelona from a geostationary satellite working in the S-band.

To begin with, note that the propagation loss J(v) due to a building edge can be roughly estimated using the knife edge diffraction model [44,45]:(2)$$J\left(v\right)=6.9+20\log\left(\sqrt{{\left(v-0.1\right)}^{2}+1}+v-0.1\right)\;\;\;\mathrm{dB}$$
where $v=\sqrt{2}\;h/b_{1}$, *h* is the Fresnel clearance distance (h<0 if the Line-Of-Sight is not blocked by the building), see Figure 3, and $b_{1}$ is the cross-sectional radius of the first Fresnel zone. Equation (Equation 2) is valid for v>−0.78 and f>30 MHz. As we can see in Figure 3, attenuation increases rapidly from 6 dB, corresponding to a zero Fresnel zone clearance situation. The high attenuation caused by building diffraction reinforces the validity of the ON-OFF (LOS-NLOS) approach in urban areas.

Figure 4 depicts a generic scenario of a street and several buildings of different heights blocking the Line-Of-Sight from the satellite. We define the “shadow zone” as the area projected over the street that establishes the boundary between LOS and NLOS situations. This approach corresponds to a zero Fresnel zone clearance situation, similar to the MKFs [33], meaning that the distance between the building and the Line-Of-Sight from the user to the satellite is zero.

The “shadow distance”, $d_{s}$, measured along the normal to the building, can be calculated by
(3)$$d_{s}=\frac{h_{b}-h_{m}}{\tan\theta }\sin\varphi$$
where $-180^{\circ }<\varphi <180^{\circ }$ is the azimuth angle of the street relative to satellite LOS direction, θ is the elevation angle, and $h_{b}$ and $h_{m}$ are the building and mobile heights, respectively. The optimal situation is when the street is parallel to the satellite Line-Of-Sight ($\varphi =0^{\circ },\pm 180^{\circ }$), resulting in a zero shadow distance (see Figure 5). On the other hand, the worst scenario occurs when the street is perpendicular to the satellite line-of-sight ($\varphi =\pm 90^{\circ }$), as the shadow distance becomes maximum in this case.

If we take 3.5 m as the typical height of the storey of each building [46] and, to fix ideas, 1.5 m as the typical height of a mobile terminal above ground level, we can approximately calculate the shadow distance $d_{s}$ as a function of the number of storeys of a building; the satellite elevation angle, θ; and the street orientation, φ, as it is depicted in Figure 6. Observe that the larger the elevation angle, the smaller the shadow zone. For LEO satellites, the elevation angle θ changes with time: it starts out very small, and then increases until the satellite is almost vertical to the ground. Then, it decreases until the satellite disappears behind the horizon. The statistical distribution of the angles is approximately exponential [47]. The combination of the elevation angle distribution and shadow lengths readily provides the distribution of the ON/OFF states for a given street width. Complementary, Figure 7 represents the shadow distance as a function of the number of storeys for different elevation angles in the worst situation (street perpendicular to satellite direction). For small elevation angles ($\theta <20^{\circ }$) the shadow distance becomes really large even for low buildings, which would probably result in a total blockage situation in that location.

## 4. Experimental Validation

Following the procedure described above, we designed a MATLAB-based software tool to calculate the shadow zones in urban environments. We tested the tool with experimental data obtained from a geostationary satellite working in the S-band (2187.5 MHz) and located at $10^{\circ }$ E. The signal transmitted was a Digital Video Broadcasting Satellite to Handheld (DVB-SH) signal [48]. Its main parameters were Coded Orthogonal Frequency Division Multiplexing (COFDM), with 1/4 guard interval, 2048 point FFT, 1705 carriers, and 5 MHz bandwidth. In addition, 16-QAM constellations were employed. The receiver was based on a development board with Xilinx FPGA. It estimated the C/N using the continuous pilots and averaging the result over several OFDM symbols to increase the accuracy of the measurement. There is a clear justification for this choice: though DVB-S broadcast technologies were primarily designed to transmit TV signals, they can effectively carry data for any purpose using the open IP over DVB protocol. Therefore, it has been proposed (e.g., see [8]) that the communication between IoT gateways and the satellite could be based on existing DVB-S (Digital Video Broadcasting by Satellite) standards for the downlink, i.e., the link from the satellite to the ground devices, and on the Digital Video Broadcasting–Return Channel Satellite (DVB-RCS) standard or the terrestrial network for the return link, i.e., the link from the ground devices to the Internet. Therefore, our testbed reproduces mainly the case of a mobile gateway that communicates with the satellite using a DVB-SH receiver.

The measures were taken in the city of Barcelona, along the route shown in the Figure 8. Dots in green color represent the locations where the carrier-to-noise ratio (C/N) exceeded the LOS level; otherwise, dots are marked in red. Taking into account the scale used in the figure, the proximity of measurement locations causes an apparent overlap between red and green dots. A receiving equipment was installed in a car that was also equipped with GPS to record the coordinates at which the measurements were made. The movement of the car was kept as continuous as possible, being limited by the presence of other vehicles, traffic lights, and traffic signs. Moreover, due to traffic conditions and the different widths of the streets, the car did not always drive at the same distance from the surrounding buildings. For this reason, even on parallel streets, measurements can vary greatly. The histogram of these experimental measures (shown in Figure 9) strongly suggests a *bimodal distribution*, supporting the hypothesis of two states or modes. The cartographic information was obtained in shapefile (SHP) format from the web map service (WMS) of the Spanish cadastre [49].

In Figure 10, Figure 11 and Figure 12 we show some examples. Locations at which the experimental values were above the LOS value have been represented in green color. The number of storeys of each building was originally coded in Roman numerals and the conversion to decimal had to be done previously. For each building on the map, its height is identified to calculate the shadow distance $d_{s}$ using Equation (Equation 3). We considered 3.5 m as the standard height of each storey [46]. For comparison, it is also shown that the proposed method clearly outperforms the coverage prediction based on masking functions (MKFs) [27,33].

Observe that there is an excellent match between the locations where there is no coverage and the shadow zones associated to the buildings. There are some places at which there is no shadow though experimental measures show no coverage. This may be because we have not considered the presence of other obstacles such as trees, lampposts, other moving vehicles, etc., that may cause signal fading. On the other hand, we have used a standard height (3.5 m) for the storeys and this is only an approximation. Moreover, on the roofs of almost every building we can find antennas, chimneys, etc. that we are not included in the cartographic data and may affect the signal propagation. In any case, these deviations are almost negligible when we consider the whole set of measurements.

Finally, in Figure 12 we show the results obtained for a configuration that does not match any of the standard scenarios considered by the MKFs defined in [27,33]. The predicted NLOS zone shows a strong coincidence with measurements that were below the LOS level.

Notice also that the model (Equation 2) holds true in the entire range of the electromagnetic spectrum under consideration and, therefore, can be used to extrapolate the experimental measurements to other frequencies. To this end, first note that the formula (Equation 2) for the propagation loss in dB due to a building edge can be asymptotically approximated by [50]
$$J\left(\nu \right)=k-10\log_{10}\left(\lambda \right),$$
where λ is the wavelength of the transmitted signal and *k* is a constant that depends on the distance between the terminal and the top of the building facade. It follows from the above formula that
$$\Delta J\approx \frac{dJ}{d\lambda }\Delta \lambda \approx -4.34\frac{\Delta \lambda }{\lambda }.$$

This equation allows us to make first-order corrections to the experimental measurements that serve to predict the attenuation *J* at other frequencies. Invoking this formula, we obtain, for example, at $f_{1}=1616$ MHz (L band), used in Iridium’s Short Burst Data service [51], that the attenuation is only 1.5 dB lower than at $f_{2}=2187.5$ MHz (the frequency used at our experiments). As the C/N (carrier-to-noise) ratio relative to LOS is larger than −1.5 dB in most of the shadowed area (see Figure 9, top), applying this slight correction to our experimental results does not alter the validity of the conclusions. Furthermore, for frequencies greater than $f_{2}$ (i.e., in the C and Ku bands), the attenuation in the shadowed area will be even greater (between 3 and 4 dB) than what we have measured, so that our general conclusions still hold true. This allows to sustain the model in the whole frequency range planned for IoT hybrid architectures. As a further comment, observe that the proposed approach deals with the C/N ratio relative to LOS, implying that other factors, such as the atmospheric attenuation, cancel out as they affect the NLOS and the LOS received power in the same way.

## 5. Discussion

In several IoT architectures, such as LoRaWAN, end devices (typically sensors) send their messages through gateways. These forward the data to a remote server, in charge of processing the information, over a satellite backhaul with a high transmission capacity (thus offloading the terrestrial 5G network). However, as shown in the above experiments (see Figure 8, Figure 9, Figure 10, Figure 11 and Figure 12), a clear line-of-sight to the satellite does not always exist or, worse yet, can be unstable when the ground-terminal moves, as occurs in *SatCom on-the-move* applications, e.g., when the satellite communicates to a drone acting as a mobile gateway [10]. As a consequence, existing IoT communication protocols tend to be inadequate and cannot be directly employed in communications involving satellite radiolinks.

To overcome the previous drawback, protocols need to be robustified against not only the roundtrip latency of the satellite, but also to the much longer propagation delays caused by disconnections. In particular, gateways cannot be mere bidirectional relays between the IoT device and the remote server. On the contrary, they should be also be able to generate autonomously some messages, such as acknowledgments (ACKs), and sending them back to the devices. For example, in LoRaWAN class A mode, after transmitting a packet to the server, the end-device waits for a predefined time (specifically, it opens two consecutive receive windows) to the response of the server [52]. If the gateway cannot relay the packet from the remote server during either of these two intervals, because the satellite link is interrupted, then it could take the initiative and send an appropriate message to the end-device, concealing the disconnection from it.

Furthermore, it also follows that the gateway should have the capacity to store the data received from the devices while moving outside of the satellite’s communication range to prevent data loss (and vice versa, messages from the satellite must also be stored until it is possible to transmit them to the device, e.g., when the receive windows are open). In the end, the satellite IoT gateway should be endowed with store-and-forward characteristics to overcome the link interruptions and provide end-to-end reliability. In this regard, the communications between the gateway and the satellite should follow any of the existing protocols for intermittent connections, depending on the application (see a review in [53,54]). The proposed model for calculating the shadow areas enables us to evaluate the protocols through extensive computer simulations in real urban conditions. Furthermore, the geometry-based algorithm may be implemented in mobile gateways to plan their path (see Figure 13).

## 6. Conclusions

We have presented a simple approach to predict the blockage areas for satellite signal reception in an urban area. Even though the model is very simple, as we are considering only the signal blockage due to buildings, the predicted values agree very well with experimental measures. This result suggest that the contributions of other obstacles as lamposts, trees, etc. are not strong enough to ensure the coverage in practice.

The proposed approach has a low computational load. This is very desirable for the purposes of generating coverage maps that can be used for the optimal allocation of the network resources.

Our future work is oriented to minimize the user intervention in the process. Furthermore, the proposed approach will be used for evaluating the performance of communication protocols for satellite-based IoT by simulations in real environments, specially in SatComm On-The-Move scenarios.

## Figures and Tables

**Figure 1 sensors-20-00475-f001:**
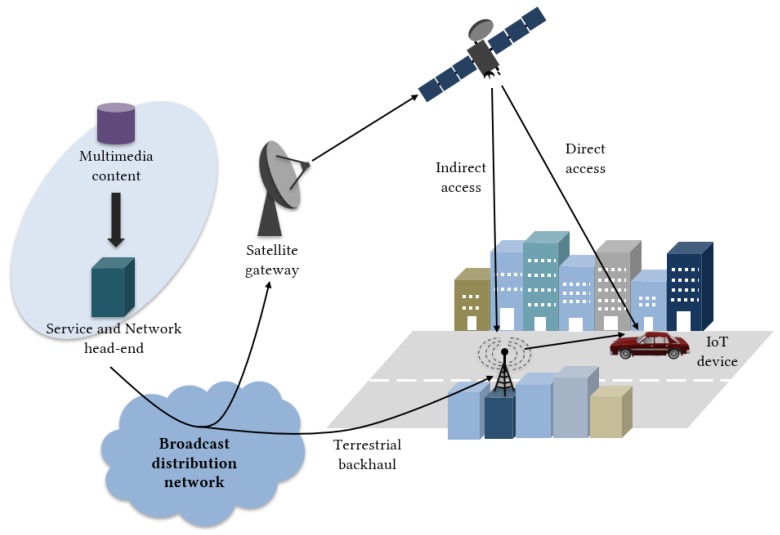
Example of integrated terrestrial-satellite network architecture.

**Figure 2 sensors-20-00475-f002:**
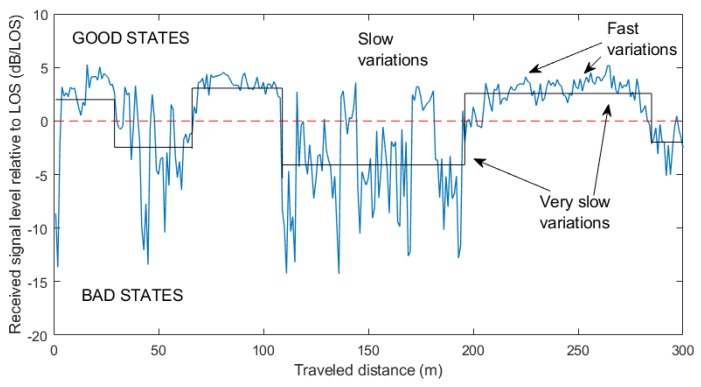
Measured signal level (S-Band) relative to LOS in an urban environment. Very slow variations are modeled by two states: GOOD and BAD.

**Figure 3 sensors-20-00475-f003:**
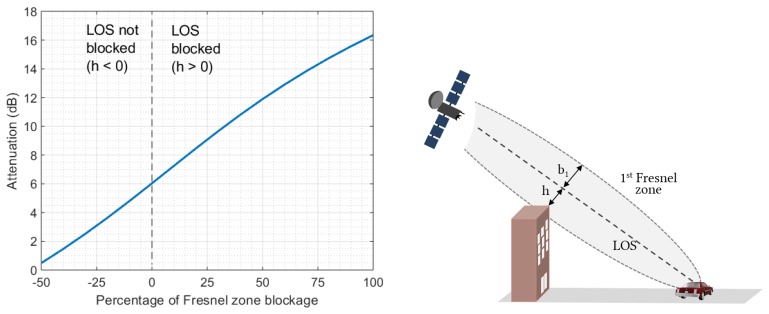
**Left**: Attenuation of the signal due to building diffraction as a function of the percentage of the first Fresnel zone obstruction. Negative values correspond to a situation where LOS is not blocked. **Right**: Depiction of Fresnel clearance distance *h* and the cross-sectional radius of the first Fresnel zone $b_{1}$.

**Figure 4 sensors-20-00475-f004:**
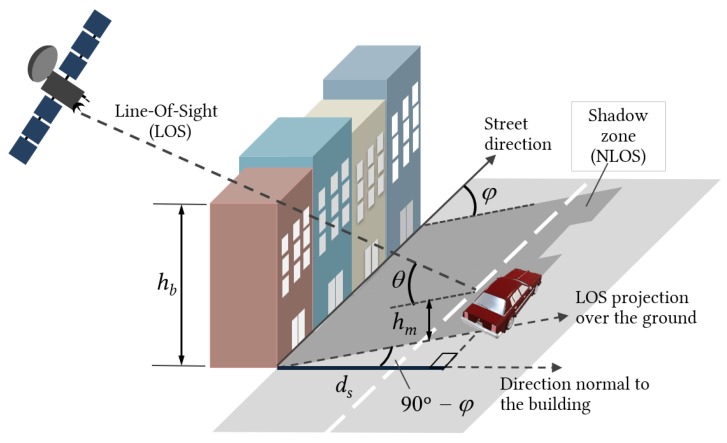
Characterization of the shadow zone. The shadow length, $l_{s}$, is measured along the LOS direction and depends on the building height, $h_{b}$. $d_{s}$ is the distance from the shadow boundary to the building, measured perpendicularly to the facade. φ is the azimuth angle of the LOS direction relative to street direction, θ is the elevation angle, and $h_{m}$ is the mobile height.

**Figure 5 sensors-20-00475-f005:**
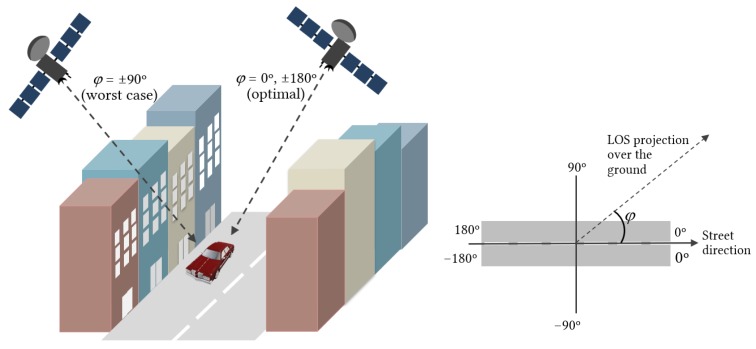
**Left**: Best and worst case scenario for satellite signal reception. **Right**: Satellite azimuth angle relative to street direction.

**Figure 6 sensors-20-00475-f006:**
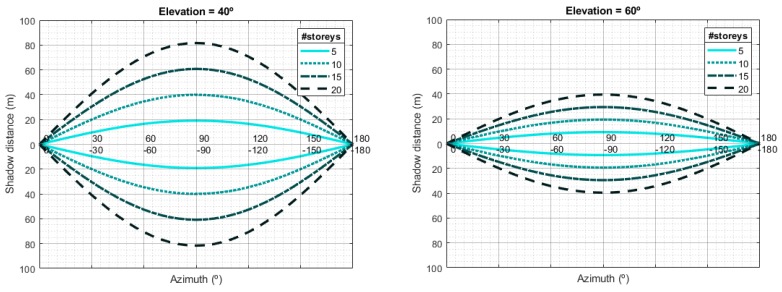
Example of shadow distance calculation as a function of the number of storeys and the street orientation (azimuth angle of the street relative to satellite LOS direction) considering two different elevation angles of the satellite.

**Figure 7 sensors-20-00475-f007:**
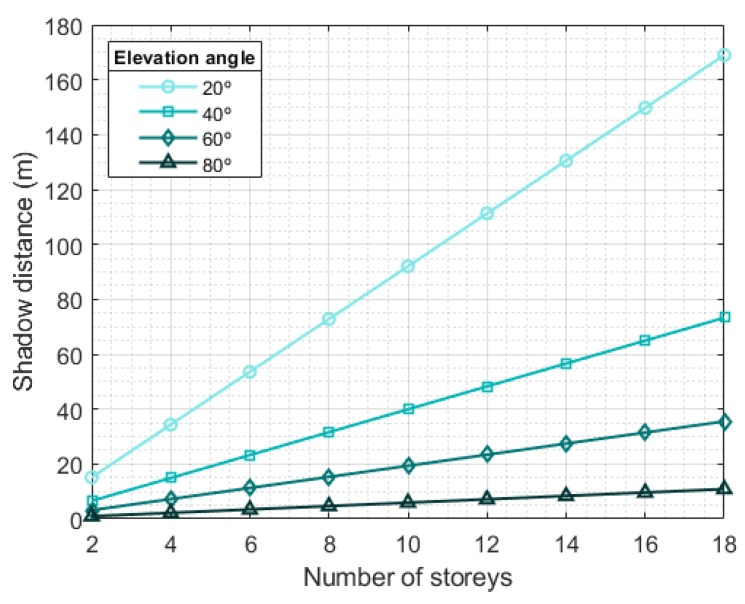
Shadow distance as a function of the number of storeys of the buildings and the satellite elevation angle for the case where the orientation of the street is perpendicular to the satellite Line-Of-Sight (worst situation).

**Figure 8 sensors-20-00475-f008:**
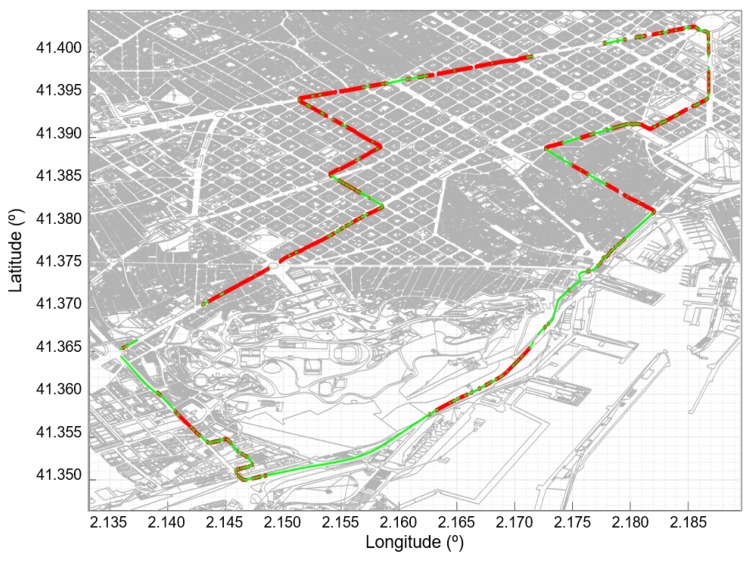
Path along which the measurements were taken, corresponding to the urban area of Barcelona. The locations where the signal exceeded/did not exceed the LOS level are shown in green/red color, respectively. There are some fragments of the path that appear without any mark because the measurements were unreliable. The apparent overlap of red and green dots is due to the scale of representation and the proximity of the measurement locations.

**Figure 9 sensors-20-00475-f009:**
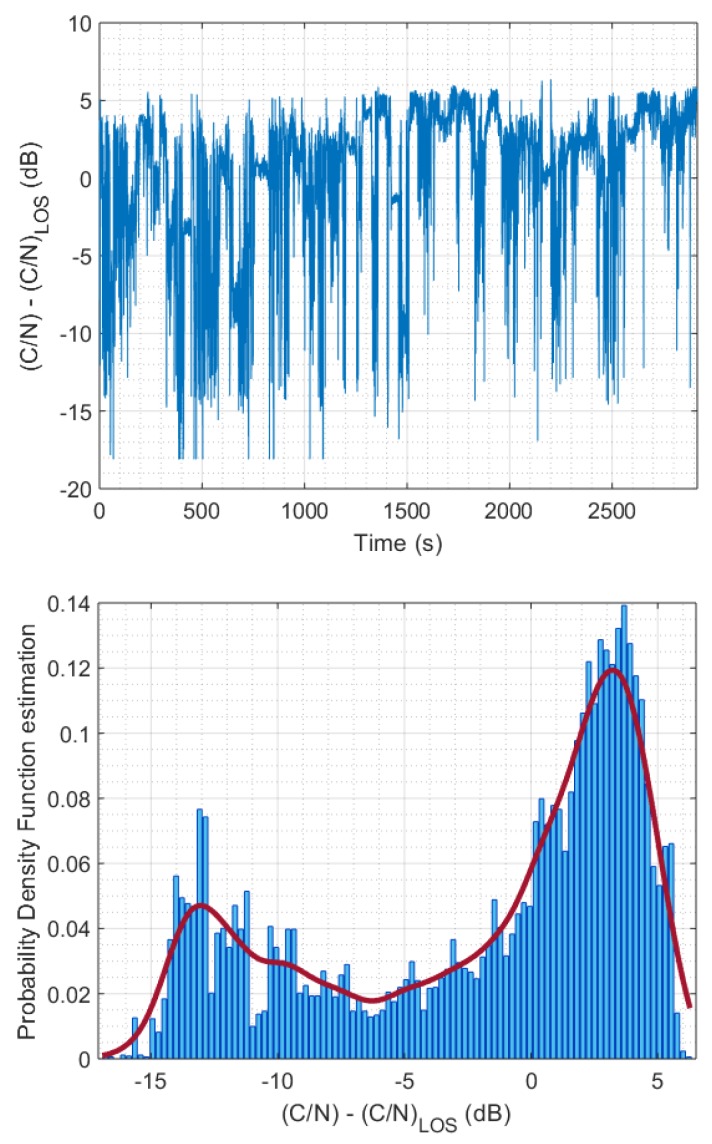
Measured carrier-to-noise ratio (C/N) relative to LOS along the route in the urban area of Barcelona, as a function of time **(top**) and estimated probability density function of the data (**bottom**).

**Figure 10 sensors-20-00475-f010:**
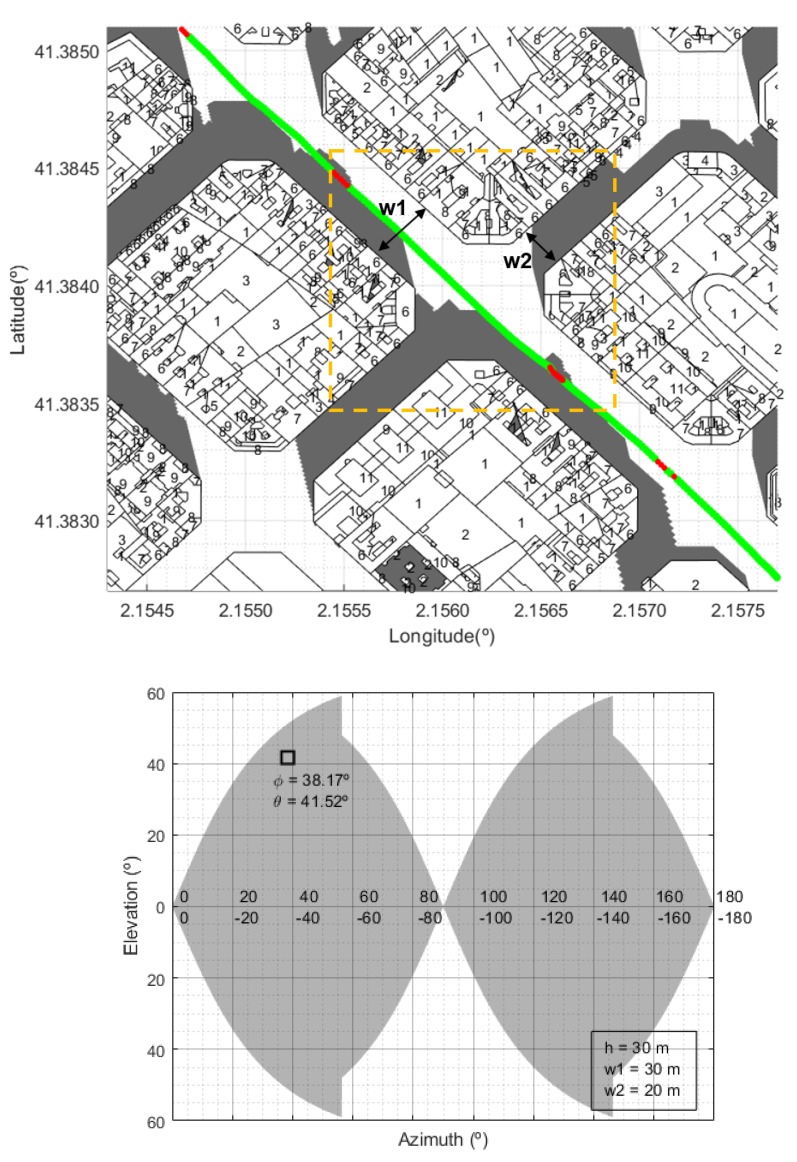
Scenario 1: Example of the shadow zones predicted for a real scenario in Barcelona where almost all the measures recorded were above the LOS level (green points). However, the MKF that corresponds to the street crossing bounded by the dashed line predict no coverage. h= average building height; *w*1, *w*2 = average streets widths; φ, θ are, respectively, the azimuth and elevation angles of the wider street relative to direction of the satellite.

**Figure 11 sensors-20-00475-f011:**
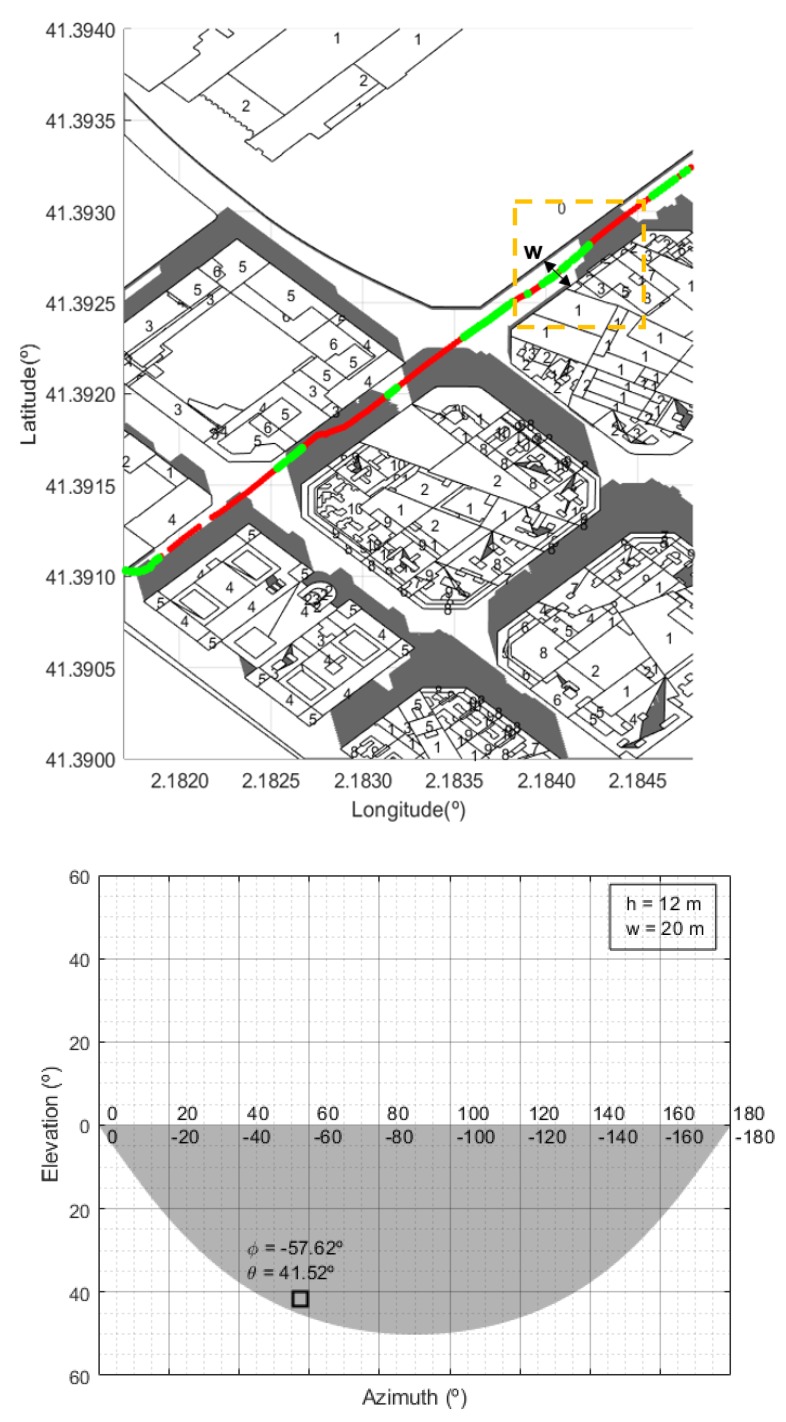
Scenario 2: Example of the shadow zones predicted for a real scenario in Barcelona. Green points represent the locations where measures are above the LOS level. The MKF that corresponds to the ”single wall” street bounded by the dashed line predict no coverage. h= average building height; w= average street width; φ, θ are, respectively, the azimuth and elevation angles of the street considered relative to the satellite.

**Figure 12 sensors-20-00475-f012:**
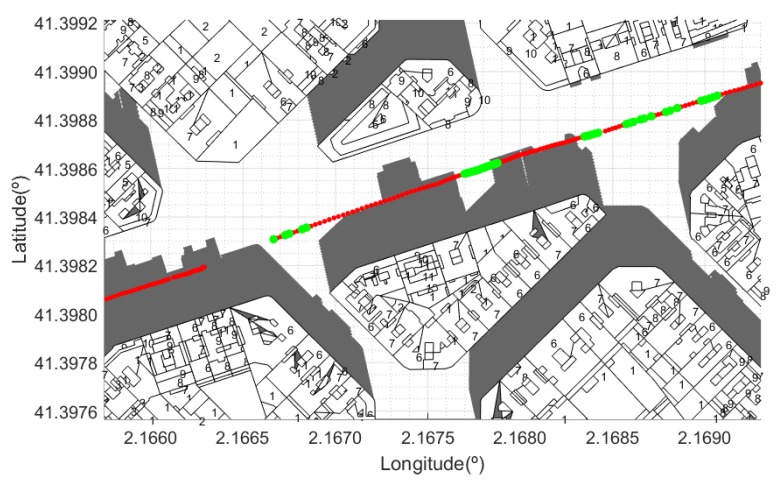
Scenario 3: Shadow zones for a street configuration that does not correspond to any of the standard MKFs. Green points refer to the locations where measures were above the LOS level.

**Figure 13 sensors-20-00475-f013:**
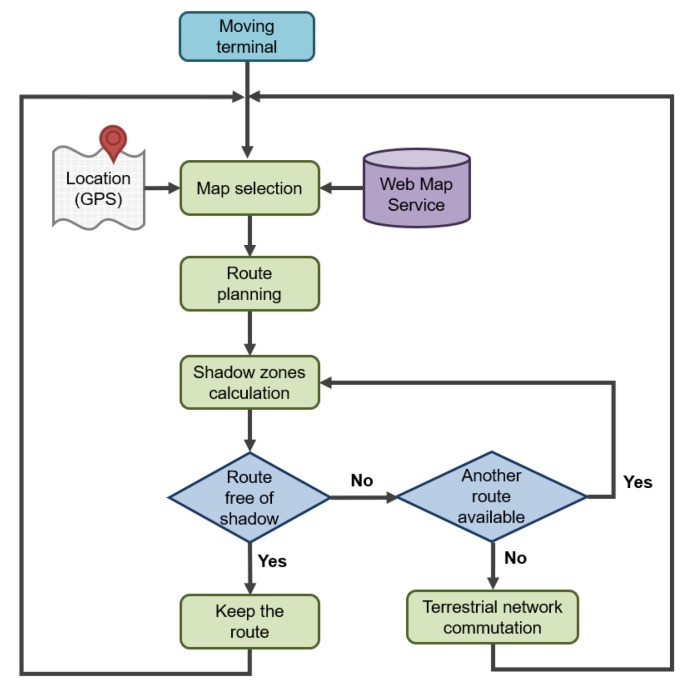
Example of flowchart representing the mobile gateway path planning procedure.

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
