# Peer review of "Prediction of Satellite Shadowing in Smart Cities with Application to IoT"

_sensors, 2020, doi:10.3390/s20020475_

Round 1

Reviewer 1 Report

This is a very interesting article that covers an area of interest to both satellite providers, HAPS operators, and those wishing to provide reliable IoT and 5G services in urban areas.

There are a few suggestions. There should be some brief discussion of the Japanese Quasi-Zenith (Figure 8) satellite configuration that has been developed to provide high angle penetration of urban environments of Tokyo, Yokohama, Osaka, etc.) in addition to LEO, MEO and GEO.

There should be some discussion of the relative effectiveness that can be achieved for various applications and the criticality of their reliability.This ranges from lower criticality for most IoT services up to the criticality of 5G services to support self-driving cars and other applications that involve life and death applications.

This articles covers the issues and the proposed technique well and these additional issues might be suggested for study and presentation in follow-on articles. I believe this is an important issue and the effective interface between Satellites, HAPS, UAVs and drones is an area there further efforts is needed and how this methodology can be used in conjunction with all these capabilities and mobile user antennas and comms on the move.

Reviewer 2 Report

This paper analyzes the impact of shadowing zones for IoT devices in the incoming 5G networks , proposes a prediction based on mapping information and validates it in the area of Barcelona.

Despite the methods are clearly described, the results are focused on a very specific area (Barcelona) and for a specific use-case (GEO). As the authors agreed, IoT & GEO is a complex and difficult combination, due to the too high delay. However, knowing that, the authors provide the results based on this premise, which seems to me not very accurated to IoT. To achieve it, they employ a car equipped by a classic receiver and they do not use any type of new IoT device. Since the title addresses IoT, I would expect a validation made by using some of this kind of receivers. 

Moreover, there is another fundamental question that is not addressed properly: the carrier frequency. The authors did the measurements in the S-band but there are some alliances whose efforts are put in the C-band. USA is pending of giving some new licenses in the C-band to terrestrial access and there are some others studies devoted to analyze the impact of this at higher frequencies. Depending on the carrier, some problems may arise and I do not detect that this aspect is commented in the paper. Furthermore, the frequency licensing and planning will be a critical aspect for hybrid architectures. I would appreciate a deeper study having in mind this issue.

Inspecting Fig. 8 there is strange phenomena. The path in Gran Via before Plaça Espanya is almost in red color, but in green ahead. Is there an explanation for this? The satellite is at W-E direction, but to me it is difficult to explain this, specially the zone behind Montjuïc mountain. Plus, why Urgell is green and Aribau is red if both are parallel?

Furthermore, there is no information on how the measurements are obtained. For instance, are they interpolated from pilots of a DVB-S2 frame? Which standard is used? Are they obtained by transmitting a known pilot all the time or similar? What bandwidth? Single carrier or multicarrier? Sampling rate? All this information for the methodology is missing.

Finally, I the authors state that "The proposed approach eliminates the necessity of site-specific information". However, they used the very specific information of Spanish cadastre. If understood correctly, they predict the coverage state (good or bad) by inspecting the height of the buildings and these predictions are validated with direct measurements taken by a car. But again, to me, performing these predictions needs having the site-specific data, such as cadastre. Definitively I do not see clearly why they state that.

In my opinion the paper is wrongly addressed. The title evokes a paper with specific IoT aspects, as well as hybrid architectures. However, the terrestrial aspect and IoT terms are circumstancial and can be perfectly removed from the text. The paper is an analysis of coverage methodology, far away from IoT & hybrid systems specific aspects. I would suggest the authors to modify the title to avoid confusing ideas and refocus the paper on what they are performing and presenting.

The proposed methodology is interesting and the validation section is realistic, but it needs a more rigorous conductive thread.

Round 2

Reviewer 2 Report

The authors addressed all the concerns with a clear and precise explanation. Therefore, I have no objections and additional comments to publish the paper as is.